# Potential Role of Lauric Acid in Milk Fat Synthesis in Chinese Holstein Cows Based on Integrated Analysis of Ruminal Microbiome and Metabolome

**DOI:** 10.3390/ani14101493

**Published:** 2024-05-17

**Authors:** Huimin Zhang, Yi Wang, Liping Hu, Jiahe Cong, Zhengzhong Xu, Xiang Chen, Shengqi Rao, Mingxun Li, Ziliang Shen, John Mauck, Juan J. Loor, Zhangping Yang, Yongjiang Mao

**Affiliations:** 1Key Laboratory for Animal Genetics, Breeding, Reproduction and Molecular Design of Jiangsu Province, College of Animal Science and Technology, Yangzhou University, Yangzhou 225009, China; minmin-911@163.com (H.Z.);; 2Jiangsu Key Lab of Zoonosis, Jiangsu Co-Innovation Center for Prevention and Control of Important Animal Infectious Diseases and Zoonoses, Yangzhou University, Yangzhou 225009, China; 3Department of Animal Sciences, Division of Nutritional Sciences, University of Illinois, Urbana, IL 61801, USA

**Keywords:** dairy cows, milk fat, rumen, microbiome, metabolome, lauric acid, bovine mammary epithelial cells

## Abstract

**Simple Summary:**

Milk fat is one of the most important economic traits in dairy cow production. Numerous investigations have demonstrated the influence of the ruminal microbiome’s composition and metabolic profile on milk production. However, no studies have directly connected the ruminal metabolome with bacterial and fungal communities as a function of milk fat content in dairy cows. The specific objective of the present study was to compare ruminal metabolome with bacterial and fungal communities among Chinese Holstein cows with contrasting milk fat content that were fed the same diet. Our results confirm that ruminal metabolome, bacterial, and fungal communities differ significantly in dairy cows with different milk fat contents. Among these different metabolites, lauric acid was enriched in fatty acid biosynthesis and selected for an in vitro study with bovine mammary epithelial cells, indicating that lauric acid promoted intracellular triglyceride synthesis via upregulation of the mRNA abundance of fatty acid uptake and activation, as well as lipogenesis. This study revealed microbial-mediated metabolic processes related to milk fat synthesis.

**Abstract:**

The composition and metabolic profile of the ruminal microbiome have an impact on milk composition. To unravel the ruminal microbiome and metabolome affecting milk fat synthesis in dairy cows, 16S rRNA and internal transcribed spacer (ITS) gene sequencing, as well as ultra-high performance liquid chromatography-mass spectrometry (UHPLC-MS/MS) methods were used to investigate the significant differences in ruminal bacterial and fungal communities as well as metabolome among Chinese Holstein cows with contrasting milk fat contents under the same diet (H-MF 5.82 ± 0.41% vs. L-MF 3.60 ± 0.12%). Another objective was to culture bovine mammary epithelial cells (BMECs) to assess the effect of metabolites on lipid metabolism. Results showed that the acetate-to-propionate ratio and xylanase activity in ruminal fluid were both higher in H-MF. Microbiome sequencing identified 10 types of bacteria and four types of fungi differently abundant at the genus level. Metabolomics analysis indicated 11 different ruminal metabolites between the two groups, the majority of which were lipids and organic acids. Among these, lauric acid (LA) was enriched in fatty acid biosynthesis with its concentration in milk fat of H-MF cows being greater (217 vs. 156 mg per 100 g milk), thus, it was selected for an in vitro study with BMECs. Exogenous LA led to a marked increase in intracellular triglyceride (TG) content and lipid droplet formation, and it upregulated the mRNA abundance of fatty acid uptake and activation (*CD36* and *ACSL1*), TG synthesis (*DGAT1*, *DGAT2* and *GPAM*), and transcriptional regulation (*SREBP1*) genes. Taken together, the greater relative abundance of xylan-fermenting bacteria and fungi, and lower abundance of bacteria suppressing short-chain fatty acid-producing bacteria or participating in fatty acid hydrogenation altered lipids and organic acids in the rumen of dairy cows. In BMECs, LA altered the expression of genes involved in lipid metabolism in mammary cells, ultimately promoting milk fat synthesis. Thus, it appears that this fatty acid plays a key role in milk fat synthesis.

## 1. Introduction

Milk fat content is an essential indicator of milk quality and performance of dairy cows and is also consistently associated with the economic interests of dairy farms [1]. It is well-established that milk fat content is affected by breed, nutrition, and stage of lactation [2]. Even under the same diet and management system, the milk fat content of Holstein dairy cows can range between 3.0% and 5.0% [3]. Milk fat consists primarily of triglycerides (TG), which include a glycerol backbone and three ester-linked fatty acids [4]. De novo synthesized fatty acids from acetate and β-hydroxybutyrate in mammary epithelial cells include C4 to C10, while C18 and higher in milk are derived either from the diet or the circulation. Fatty acids with C12–C16 have a dual origin. Approximately 30% of C12:0 in milk fat arises from plasma TG, whereas uptake from the blood of C14:0 and C16:0 account for 70% and 80% of the total in milk fat [5]. The ruminal microbiome is responsible for the production of acetate, butyrate, and other precursors or components (e.g., odd-chain fatty acids) of milk fat [6]. The key contribution of the microbiome to milk fat synthesis was highlighted recently by studies demonstrating that the composition and metabolic profile of the microbiome impact milk production [2,7].

Recently, 16S rRNA gene sequencing has been used to profile the bacterial communities in the rumen, revealing that three genera of the family *Lachnospiraceae* and four genera of the order *Bacteroidales* were the best biomarkers for milk fat content [2]. *Cutaneotrichosporon* and *Cyphellophora*, classified as fungi, have been found in greater abundance in the rumen of cows that produced milk with higher concentrations of saturated fatty acids (SFA) [8]. Several studies of diet-induced milk fat depression revealed that ruminal bacteria involved in biohydrogenation could be the main factor affecting milk fat synthesis via the production of *trans*-10, *cis*-12 conjugated linoleic acid [7,9].

High-throughput metabolomics can identify many metabolites from samples and help uncover metabolic phenotypes in organisms [10]. Many studies based on metabolomics technology have uncovered metabolic biomarkers and pathways related to milk fat content in response to diet or breed; thus, this technology provides an opportunity to better understand the mechanism of milk fat synthesis [11,12]. A recent study comparing the gut metagenome and metabolome of Zhongdian yak cows with different milk fat content underscored the existence of positive correlations between bacteria and metabolites (e.g., myristic acid and choline) with the inhibition of methanogenesis. These data led authors to speculate that gut bacteria in yak cows with greater milk fat content were metabolically more efficient at generating energy-yielding metabolites for use by the mammary gland for milk fat synthesis [13].

Most studies investigating the ruminal microbiome related to milk fat content have focused primarily on bacteria, the most abundant and diverse members (10^10^–10^11^ cells per gram) [14]. Fungi, despite their low numbers (10^3^–10^5^ cells per gram), play a significant role in the degradation of plant fiber by producing a vast array of polysaccharide-degrading enzymes, including cellulase, xylanase, and other hydrolytic enzymes [14,15]. Compared with bacteria, little research related to milk fat content has focused on the role of fungi. Further, no studies have specifically linked the ruminal metabolome with bacterial and fungal communities as a function of milk fat content in dairy cows.

Various studies have reported a correlation between milk fat content and ruminal microbiome and their metabolome [2,7,12]. Based on the previous studies, it was hypothesized that certain ruminal bacteria or fungi could produce some metabolites that might reach to mammary glands through blood circulation, and further affect milk fat synthesis in dairy cows. To test our hypothesis, we compared the ruminal metabolome with bacterial and fungal communities among Chinese Holstein cows with contrasting milk fat content but fed the same diet. Then the effect of the key metabolite (lauric acid, LA) on TG synthesis was explicated in vitro study with bovine mammary epithelial cells (BMECs).

## 2. Materials and Methods

### 2.1. Cows, Experimental Design, and Diets

Based on 2020 DHI data (from February 2020 to January 2021), 5 Chinese Holstein cows with chronically low milk fat content (3.60 ± 0.12%; L-MF group) and 5 cows with chronically high milk fat content (5.82 ± 0.41%; H-MF group) were selected from 532 cows in Wanhe dairy farm (Xuzhou, China). As shown in Appendix A, the mean milk fat content of H-MF group always tended to be higher than L-MF group during summer, autumn, and winter in 2020 (*p* ≤ 0.1). The L-MF and H-MF cows are half-sibs, and did not differ in parity (1.80 ± 0.37 vs. 1.80 ± 0.37), lactation days (257.80 ± 55.90 vs. 218.80 ± 77.98), milk yield (28.54 ± 1.90 vs. 27.96 ± 5.34 kg/d), and dry matter intake (22.50 ± 1.85 vs. 21.57 ± 1.80 kg/d). Cows were kept in free stall housing and fed the same total mixed ration (TMR) (Appendix A), formulated to meet or exceed NRC (2001) nutrient specifications for lactating dairy cows. Daily dry matter intake for each cow was recorded weekly for four weeks before sampling.

### 2.2. Sample Collection

Cows were milked three times daily, and individual milk yield was recorded. Pooled milk samples (morning, midday, and evening) were collected and stored at 4 °C (pharmacy refrigerator with freezer YCD-265, Aucma, Qingdao, China) and used for milk composition analysis via MilkoScan™ FT3 (Foss, Hilleroed, Denmark) in 3 days. Quantities of 10 mL milk samples were immediately frozen in liquid nitrogen and stored at −80 °C until fatty acid analysis. Mixed ruminal contents were collected from each cow via esophagus tubing (2000 mm × 18 mm stainless steel hose with a syringe) approximately 2 h after the morning feeding. Subsequently, ruminal fluid pH was measured with 50 mL ruminal contents by a mobile FiveGo F2 pH meter (Mettler Toledo, Greifensee, Switzerland). Amounts of 10 mL of ruminal contents were immediately frozen in liquid nitrogen and stored at −80 °C until further analysis.

### 2.3. VFA Concentration and Digestive Enzyme Activity in Ruminal Fluid

Frozen ruminal contents were thawed and centrifuged at 10,000× *g* for 10 min. Then, 0.2 mL of 20% metaphosphoric acid was added to 1.0 mL ruminal fluid supernatant. After centrifugation at 10,000× *g* for 10 min, quantification of VFAs was done via GC (7890-B gas chromatograph, Agilent, Santa Clara, CA, USA) equipped with a DB-WAX fused silica capillary column (30 m × 0.25 mm × 0.2 µm, Agilent). The GC operation and data analysis were conducted as described by Cao et al. [16]. The activities of cellulase, xylanase, lipase, pectinase, proteinase, and amylase in ruminal fluid were measured using ELISA commercial kits (Nanjing Jiancheng Bioengineering Institute, Nanjing, China). Briefly, the enzyme is specifically recognized both by a solid phase antibody coated on the ELISA plates and the horseradish peroxidase (HRP)-conjugated antibody. Firstly, pipet 50 µL of standards, control, and rumen fluid supernatant samples into appropriate wells in triplicate. Cover the plate and incubate at 37 °C for 30 min. Following being washed by PBST, add 50 µL of anti-enzyme HRP conjugate into each well. After incubation and washing, 100 µL of TMB color reagent solution was added to each well. Finally, the reaction was stopped by adding 50 µL of stop solution, and the optical density of the colored solution was measured at 450 nm by using a microplate reader (Thermo Fisher, Waltham, MA, USA).

### 2.4. DNA Extraction, Amplification, and Sequencing

According to manufacturer instructions, microbial DNA was extracted from ruminal digesta using the HiPure Stool DNA Kit (Magen, Guangzhou, China). Total DNA concentration was evaluated using a NanoDrop 2000 spectrophotometer (Thermo Fisher Scientific Inc., USA). For bacteria, the V3-V4 hypervariable region of 16S rRNA was amplified by PCR using primer pairs 341F (5′-CCTACGGGNGGCWGCAG-3′) and 806R (5′-GGACTACHVGGGTATCTAAT-3′). For fungi, the internal transcribed spacer 2 (ITS2) rRNA region was amplified by PCR with primer pairs ITS3_KYO2 (5′-GATGAAGAACGYAGYRAA-3′) and ITS4 (5′-TCCTCCGCTTATTGATATGC-3′). According to the manufacturer, purified amplicons were pooled in equimolar and were paired-end sequenced (PE250) on the NovaSeq 6000 System (Illumina, San Diego, CA, USA).

### 2.5. Metabolomics Analysis

An ultra-high-performance liquid chromatography-mass spectrometry (UHPLC-MS/MS) analysis was carried out using a Vanquish UHPLC system (ThermoFisher, Germany) equipped with a Hypesil Gold column (100 mm × 2.1 mm, 1.9 μm), coupled with an Orbitrap Q ExactiveTMHF-X mass spectrometer (Thermo Fisher, Dreieich, Germany) in Gene Denovo Co., Ltd. (Guangzhou, China). The injection volume was 3.0 μL, and the column temperature was maintained at 40.0 °C. Separation was achieved with eluent A and eluent B (Methanol) with the following gradient at a flow rate of 0.20 mL min^−1^: 2% B with a 1.5 min hold; 2–100% B over 1.5–12.0 min; 100% B with a 2.0 min hold; 100–2% B over 14.0–14.1 min; and 14.1–17.0 min holding at 2% B. Eluent A (0.1% formic acid in water) was used in the positive polarity (POS) mode, and eluent A (5 mM ammonium acetate, pH 9.0) was used in the negative polarity (NEG) mode. The mass spectrometric data were collected using a Q Exactive^TM^ HF-X mass spectrometer, which was operated in POS/NEG mode with a spray voltage of 3.2 kV, capillary temperature of 320 °C, sheath gas flow rate of 40 arb, and aux gas flow rate of 10 arb. Centroid data were collected from 100 to 1500 *m*/*z*.

### 2.6. Milk LA Analysis

Milk fatty acid extraction and analysis were conducted as described in our previous study [17]. Briefly, milk fat was isolated by centrifugation, and 50 mg of milk fat was used for methylation. The fatty acid methyl esters were identified using an Agilent 7890A gas chromatography system (Agilent Technologies, Santa Clara, CA, USA) equipped with a DB-HP capillary column (100 m × 0.25 mm × 0.2 μm, Agilent, USA). C11:0 (1 mg mL^−1^, Sigma, St. Louis, MO, USA) was added as an internal standard. The retention time of each fatty acid was verified based on the peaks of a Supelco 37 fatty acid methyl ester standard mix (Sigma, USA). The concentration of milk fat LA (mg per 100 g milk) was calculated according to the internal standard and milk fat content.

### 2.7. Microbiome and Metabolome Data Analysis

The microbial sequencing data were analyzed using the following procedures. Briefly, the raw reads were filtered using FASTP (version 0.18.0) to get high-quality clean reads, which were further merged and filtered to obtain the high-quality clean tags using FLASH (version 1.2.11). The clean tags were then clustered into bacterial or fungal operational taxonomic units (OTUs) using the UPARSE (version 9.2.64) pipeline, with a 97% sequence similarity threshold. Taxonomic classification of the representative OTU sequences was assigned using the RDP classifier (version 2.2) based on the SILVA database (version 138.1) for bacteria and the UNITE database (version 8.3) for fungi, with a confidence threshold value of 0.8. α diversity analyses, including ACE, Chao1, Sob, Shannon, and Simpson indices, were calculated with QIIME (version 2.0). β diversity at the OTU level was visualized with principal coordinate analysis (PCoA) based on unweighted UniFrac distance calculation using the R project ggplot2 package (version 2.2.1). A stacked bar plot of the taxonomic composition was also generated in the R project ggplot2 package. α index and genus comparison between L-MF and H-MF groups were calculated by Welch’s *t*-test in the R project Vegan package (version 2.5.3). SparCC correlation analysis between ruminal bacteria and fungi at the genus level was calculated in the R project SpiecEasi package (version 1.0.7). The heatmap of genus correlations was plotted using the pheatmap package (version 1.0.12) in the R project.

The procedure of metabolome data analysis was conducted as described by Li et al. [18]. Compound Discoverer 3.1 (Thermo Fisher, USA) software was used to process raw data generated by UHPLC-MS/MS, including extraction, alignment, picking, and quantitation of the peak. After normalization of peak intensity, the molecular formula was predicted based on additive ions, molecular ion peaks, and fragment ions. Then, mzCloud, mzVault, and MassList databases were used for qualitative and relative quantitative analysis of metabolites. The processed data were assessed by principal components analysis (PCA) and partial least squares discriminant analysis (PLS-DA) using the R project gmodels package (version 2.18.1.1) and ropls package (Version 1.6.2), respectively. Based on the PLS-DA results, metabolites were plotted according to their importance in separating the milk fat groups and each metabolite received a variable importance in the projection (VIP) value. Student’s *t*-test was used to analyze the differences in metabolites. The metabolites with VIP > 1 and *p* < 0.05 were considered to be differential metabolites.

The Pearson correlation coefficients between microbiota and metabolomic datasets were calculated in R (version 3.5.1). Furthermore, the correlations were visualized using a network, which was performed using the R project igraph package.

### 2.8. Cell Culture and Treatment

Cell culture of BMECs was conducted as previously described [17]. The BMECs were incubated in the presence of various concentrations (0, 50, 100, 200, 300, and 400 μM) of LA (Sigma, Subang Jaya, Malaysia) to investigate the dose effect of LA. The cells were incubated at 37 °C in 5% CO_2_. BMECs without LA served as controls (NC).

### 2.9. Cell Viability, Proliferation, and Apoptosis Determination

The cell viability was measured using cell counting kit-8 (CCK-8; Vazyme, Nanjing, China). After BMECs treatment with different concentrations of LA for 1, 2, and 3 days in a 96-well plate, 10 μL of CCK-8 solution was added to each well, and the BMECs were incubated for three hours. The number of viable cells was assessed by measuring the absorbance at 450 nm using a microplate reader (Thermo Fisher, USA). Ethynyl deoxyuridine (EdU) incorporation assay was performed to evaluate cell proliferation. After incubation with LA for 2 days, BMECs were harvested and reseeded in 6-well plates for EdU cell proliferation assays (Beyotime Biotechnology Inc., Haimen, China), according to the manufacturer’s protocol. Briefly, BMECs were incubated with 20 μM EdU for two hours. Cells were fixed with 4% paraformaldehyde, mixed with 0.3% Triton-X-100, and then stained with DAPI staining reaction solution and Hoechst 33,342 reaction mixture under dark conditions for 30 min. The EdU-labeled cells were photographed under a Leica DMi8 inverted fluorescence microscope (Leica, Wetzlar, Germany).

After treatment with different concentrations of LA for 2 days, BMECs were stained with Annexin V-FITC/PI reagent (Vazyme, China) under dark conditions for 15 min. Subsequently, the cell apoptosis rate was determined via CyAn ADP7 flow cytometry (BECMAN, Krefeld, Germany).

### 2.10. Oil Red O Staining and Intracellular TG Content

After incubation with 200 μM LA, BMECs were rinsed twice with PBS and used for lipid analysis. According to the manufacturer’s protocols, lipid accumulation was analyzed with an oil-red O staining kit (Jiancheng, China). Briefly, cells were fixed with 4% paraformaldehyde and stained with an oil-red O staining solution for 30 min. After washing, Mayer Hematoxylin staining solution was added to stain the nucleus. Cell lipid droplets were observed under a Leica DMi8 inverted fluorescence microscope (Leica, Germany). Intracellular TG was analyzed with a tissue triglyceride assay kit (Applygen, Beijing, China) according to the manufacturer’s instructions. TG content was assessed by measuring the absorbance at 550 nm using a microplate reader (Thermo Fisher, USA). The data were normalized to the protein content determined with the Pierce^®®^ BCA protein assay kit (Thermo Scientific, USA).

### 2.11. RT-qPCR and Western Blotting

Total RNA extraction, reverse transcription into cDNA, and qPCR were conducted as previously described [17]. Primer sequences (with their respective PCR fragment lengths) are described in Appendix A. Glyceraldehyde 3-phosphate dehydrogenase (*GAPDH*) was used as the control gene to normalize the amount of cDNA in the qPCR reaction. The data were analyzed using the relative quantification (2^−ΔΔCt^) method [19].

The extraction and quantification of protein were as previously described [17]. Then, the protein was separated on SDS-PAGE, transferred to the PVDF membrane (Millipore, Burlington, MA, USA), and blocked with skimmed milk. Primary antibodies included DGAT1 Rabbit Polyclonal Antibody (AF6696, 1:1000), SREBP1 Rabbit Polyclonal Antibody (AF8055, 1:1000), and β-Actin Rabbit Monoclonal Antibody (AF5003, 1:1000). HRP-labeled Goat Anti-Rabbit IgG(H+L) (A0208, 1:1000) was used as the secondary antibody. All antibodies were purchased from Beyotime Biotechnology Inc. (China). Lastly, the proteins on the membranes were visualized with the Pierce ECL Western blotting substrate (Thermo Fisher, USA) and analyzed using a digital imager (ChemiDoc XRS+ system, Bio-Rad, Hercules, CA, USA).

### 2.12. Statistical Analysis

Statistical analysis was performed with SPSS 17.0 software. The data are expressed as the means ± standard error (SE). The normal distribution of data was confirmed by Shapiro–Wilk test. The data in our study were normally distributed. Student’s *t*-test was used to analyze the differences in milk yield and composition, ruminal VFA, and enzyme activity between L-MF and H-MF groups. In addition, the same statistical analysis was used for cell viability, proliferation and apoptosis rate, TG content, and mRNA expression in the cell culture study. Differences were considered statistically significant when *p* ≤ 0.05 and considered a trend when 0.05 < *p* ≤ 0.10.

## 3. Results

### 3.1. Milk Yield and Composition

As shown in Table 1, there was no significant difference in parity, lactation days, daily milk yield, and dry matter intake between both groups (*p* > 0.05). The average milk fat content (from February 2020 to January 2021) was 3.51 ± 0.12% and 4.55 ± 0.19% in L-MF and H-MF cows. Compared with the L-MF group, the average milk fat content was greater in the H-MF group (*p* < 0.01). A similar result was observed on the sampling day; the H-MF group had greater milk fat content (5.82 ± 0.41%) (*p* < 0.01) and whole milk solids content (16.04 ± 0.86%) (*p* < 0.05) than L-MF group (3.60 ± 0.12%; 12.98 ± 0.58%). No significant differences were observed in milk protein, lactose, solids-not-fat content, milk urea nitrogen (MUN) content, and somatic cell score (SCS) between H-MF and L-MF groups on sampling day (*p* > 0.05).

### 3.2. Ruminal Fermentation Profile and Enzyme Activity

The pH of ruminal fluid was not significantly different between L-MF and H-MF groups (*p* > 0.05, Table 2). The acetate-to-propionate ratio was greater in the H-MF group compared with the L-MF group (*p* < 0.05). There was no difference in acetate, propionate, isobutyrate, butyrate, isovalerate, valerate, and total VFA concentrations between both groups (*p* > 0.05). Compared with the L-MF group, xylanase activity in ruminal fluid was greater (*p* < 0.05), whereas lipase and proteinase activities were lower (*p* < 0.05) in the H-MF group. Amylase activity tended to be greater in the H-MF than L-MF group (*p* = 0.08). In addition, cellulase and pectinase activities did not differ between the groups (*p* > 0.05).

### 3.3. Ruminal Bacterial and Fungal Communities

A total of 1,287,775 raw reads were obtained for bacterial 16S rRNA genes. After combining, trimming, and aligning, 16,427 OTUs were obtained. In total, 1,261,728 raw reads were obtained for fungal ITS2 genes, and 1,261,083 valid reads were obtained after screening. Then, these reads were binned into 2250 OTUs. Analysis of the α-diversity of microbes in ruminal fluid samples was conducted after normalization with the QIIME platform (Table 3). For bacteria, no difference was detected between L-MF and H-MF groups in the α diversity indices of microbial communities (*p* > 0.05). For fungi, the index of Chao1 and Ace revealed a trend being greater in the L-MF than in the H-MF group (*p* = 0.09, *p* = 0.08). PCoA was used to evaluate differences between the samples according to the matrix of β diversity distances. The results indicated that the two groups were largely separated from each other at the OTU level, either in bacteria or fungi (Figure 1).

### 3.4. Variation in the Ruminal Bacterial and Fungal Composition

Twenty-two bacterial phyla were identified across all samples, with *Firmicutes* and *Bacteroidetes* being the most abundant (Figure 2A). The ratio of *Firmicutes* to *Bacteroidetes* in the H-MF group was 1.04, greater than the L-MF group (0.97). At the genus level, the bacterial community was dominated by *Prevotella_1* (20.66%), *Succiniclasticum* (15.69%), and *Succinibrionaceae_UCG-001* (6.02%). There were five fungal phyla identified in ruminal samples (Figure 2B). *Ascomycota* and *Neocallimastigomycota* were particularly dominant at the phylum level, and the most dominant fungal genus was *Dipodascus*, followed by *Mycosphaerella* and *Candida*.

The relative abundance of the bacterial and fungal genera between L-MF and H-MF cows is reported in Appendix A. For bacteria (Figure 3A), the abundance of *Christensenellaceae_R-7*, *Eubacterium_coprostanoligenes*, *Ruminococcaceae_UCG-010*, *Butyrivibrio_2*, *Methanobrevibacter*, *p-1088-a5_gut*, *Acetitomaculum*, *Lachnoclostridium_10,* and *Lachnospiraceae_ND3007* was lower in the H-MF than L-MF group (*p* < 0.05). In addition, the H-MF cows had a greater abundance of *Eubacterium_xylanophilum* (*p* < 0.05). For fungi (Figure 3B), the H-MF cows had a greater abundance of *Pyrenophora*, with a lower abundance of *Caecomyces*, *Wallemia,* and *Pithoascus* than L-MF cows (*p* < 0.05).

SparCC analysis using genus-level information was conducted to identify the correlation between ruminal bacteria and fungi. A total of 225 significant pair-wise correlations (|r| > 0.5 and *p* < 0.05) were obtained. Of these, there were 69 positive and 156 negative correlations (Appendix A). The key bacterial genus and associated key fungal interactions are reported in Figure 3C; eight bacterial genera, including *Ruminococcaceae_NK4A214*, *Rikenellaceae_RC9_gut*, *Papillibacter, Ruminococcaceae_UCG-010*, *Anaerovorax*, *p-1088-a5_gut*, *Christensenellaceae_R-7,* and *Eubacterium_coprostanoligenes* were negatively correlated with *Wickerhamomyces*, *Candida,* and *Pyrenophora*. Notably, *Pyrenophora* was significantly enriched in the H-MF group. The fungal genera *Orpinomyces* was positively correlated with six bacterial genera, such as *Ruminococcaceae_UCG-010* and *Butyrivibrio_2*, significantly enriched in the L-MF group.

### 3.5. Ruminal Fluid Metabolite and Milk LA Analysis

A total of 1577 metabolites in positive ion mode and 990 metabolites in negative ion mode were initially identified. Among them, 96 metabolites differed significantly between L-MF and H-MF groups based on the *t*-test (fold change ≠ 1 and *p* < 0.05). These were mainly fatty acyls, organooxygen compounds, prenol lipids, steroids and steroid derivatives, benzene, and substituted derivatives (Appendix A). PCA and PLS-DA were conducted to characterize variations in the metabolic profiles of L-MF and H-MF cows. As reported in Figure 4, the L-MF and H-MF groups were clearly separated in the PCA plot based on metabolite data. In the PLS-DA analysis, the R2X, R2Y, and Q2Y parameters were greater than 0.5, indicating that the model yielded stable and accurate predictions.

Considering the results of the statistical analysis and the VIP value obtained from PLS-DA, 11 significantly different metabolites (*p* < 0.05 and VIP > 1) were identified (Table 4), and contributed most significantly to the separation of ruminal samples between L-MF and H-MF cows. Compared with the L-MF group, the levels of three fatty acyls (lauric acid ethyl ester, octadecanedioic acid, and LA), two lipids (12,13-dihydroxy-9(Z)-octadecenoic acid ‘12(13)-DiHOME’ and branched fatty acid esters of hydroxy fatty acid ‘FAHFA (16:0/18:2)’), and one carboxylic acid and derivative (succinic acid) were greater in the H-MF group, while phosphatidylcholine (PC (18:2/19:2)), acetophenone, and 4-(diethylamino) salicylaldehyde were lower. The KEGG pathway analysis of these metabolites revealed that acetophenone, succinic acid, LA, and phylloquinone were enriched in various physiological and biological functions (Appendix A). In particular, LA was enriched in fatty acid biosynthesis (*p* < 0.1), which is related to lipid metabolism. Succinic acid was enriched in propanoate metabolism and butanoate metabolism (*p* < 0.05), which are related to carbohydrate metabolism.

As shown in Table 1, the average concentration of LA isolated from milk fat of the H-MF group (217.13 mg per 100g milk) was greater compared with the L-MF group (156.48 mg per 100 g milk) (*p* < 0.01).

### 3.6. Correlation Analysis of Differential Metabolites and Microbiota

Correlations between differential metabolites and microbiota were explored using Pearson correlation analysis. A total of 66 significant pair-wise correlations (|r| > 0.6, *p* < 0.05) between metabolite and microbiota sets were obtained (Figure 5, Appendix A). There were 50 positive and 16 negative correlations between differentially affected metabolites and microbiota. In general, bacteria had 42 pair-wise associations with differentially affected metabolites, and fungi had 24 pair-wise associations. The differentially affected bacterial genus *Acetitomaculum* was positively correlated with acetophenone and PC (18:2/19:2), but negatively correlated with succinic acid. *Lachnospiraceae_ND3007* was positively correlated with PC (18:2/19:2), while negatively correlated with FAHFA (16:0/18:2). *Eubacterium_xylanophilum* was positively correlated with FAHFA (16:0/18:2), while negatively correlated with acetophenone.

The differentially affected fungal genus *Caecomyces* had negative correlations with LA, FAHFA (16:0/18:2), and octadecanedioic acid, but had a positive correlation with PC (18:2/19:2). However, *Pyrenophora* exhibited a reverse relationship, which showed positive correlations with LA, FAHFA (16:0/18:2), octadecanedioic acid, succinic acid, lauric acid ethyl ester, and amitriptyline-d3. In contrast, it was negatively correlated with PC (18:2/19:2) and acetophenone. The correlation network revealed that nearly all differentially affected metabolites between L-MF and H-MF had relationships with the abundance of *Pyrenophora, Caecomyces,* and *Hymenula*.

### 3.7. Effect of LA on Cell Proliferation Activity and Apoptosis of BMECs

As reported in Figure 6A, the cell proliferation of BMECs was affected by time, LA concentration, and time × LA concentration (*p* < 0.01). The BMEC proliferation gradually decreased with the extension of incubation time. The relative cell viability of LA-treated BMECs for 2 days was 1.03, similar to the NC group, but lower than 1 day (1.16) and higher than 3 days (0.92) (*p* < 0.01). A quantity of 50 and 100 μM LA significantly stimulated cell proliferation of BMECs (*p* < 0.05), while no difference was observed between NC and higher doses of LA-treated BMECs (200 and 400 μM). A quantity of 300 μM LA significantly inhibited the BMEC proliferation (*p* < 0.05). The EdU incorporation assay revealed that the percentage of EdU positive BMECs was significantly (*p* < 0.01) decreased by treatment with 300 and 400 μM LA for 2 days (Figure 6B,C), while no difference was observed between NC and the lower dose of LA-treated BMECs (50, 100 and 200 μM).

The lower-right and upper-right quadrants of the flow cytometry analysis represent early and late apoptosis of cells. Approximately 7.48 ± 0.26% cell apoptosis (early apoptosis 2.03 ± 0.1%, and late apoptosis 5.45 ± 0.16%) was observed without LA in the NC group (Figure 7). Lower doses of LA-treated BMECs (50, 100, 200, and 300 μM) had similar apoptotic cell death compared with the NC group. While compared with the NC group, higher apoptosis was observed with 400 μM LA-treated BMECs (*p* < 0.05), its apoptotic cell death reaching 10.77 ± 3.5% (early apoptosis 2.07 ± 0.84%, and late apoptosis 8.7 ± 2.66). Thus, 200 μM LA was selected and used for subsequent studies.

### 3.8. Effect of LA on Lipid Metabolism in BMECs

As reported in Figure 8A,B, treatment with 200 μM LA for 2 days dramatically increased the lipid droplet accumulation in BMECs. The TG content was significantly greater in LA-treated BMECs compared with the NC group (Figure 8C) (*p* < 0.05).

Compared with the NC group (Figure 8D), treatment with LA in BMECs dramatically upregulated the mRNA abundance of fatty acid translocase (*CD36*), acyl-CoA synthetase long-chain family member 1 (*ACSL1*), diacylglycerol acyltransferase 1 (*DGAT1*), *DGAT2*, glycerol-3-phosphate acyltransferase (*GPAM*), and sterol regulatory element binding protein 1 (*SREBP1*), while it downregulated that of fatty acid synthetase (*FASN*) and proliferator-activated receptor gamma (*PPARG*) (*p* < 0.05). There was no significant change in the abundance of fatty acid binding protein 3 (*FABP3*), acyl-CoA synthetase short-chain family member 2 (*ACSS2*), lipin 1 (*LPIN1*), perilipin 2 (*PLIN2*), and 1-acylglycerol-3-phosphate O-acyltransferase 6 (*AGAPT6*) (*p* > 0.05). As reported in Figure 8E, the protein expression levels of DGAT1 and SREBP1 were significantly increased.

## 4. Discussion

### 4.1. Association of Milk Fat Content Divergence with Ruminal Fermentation Profiles and Biopolymer Degradation Enzyme Activity

During fermentation, ruminal microbial carbon degradation produces acetate, butyrate, and propionate, which are readily absorbed across the rumen wall. Acetate and butyrate are precursors for milk fat synthesis [20]. Despite this known link, in a review of data from 28 studies, the proportion of acetate in the rumen was not associated with milk fat content (r^2^ ≤ 0.118) [7]. Differences across studies might be partly associated with the stage of lactation of the cows at sampling (50 to 200 lactation days total in Seymour et al., and Ramirez-Ramirez et al.) [21,22]. The number of cows used in the study (e.g., six total in Wu et al., 20 total in Ramirez-Ramirez et al.) also could impact the strength of this relationship [20,21]. Thus, it was not unexpected that in the present study cows in the L-MF and H-MF group had similar acetate and butyrate concentrations in the rumen.

It was reported that the acetate-to-propionate ratio in ruminal fluid can be used as a comprehensive indicator for evaluating milk quality, and it is positively correlated with milk fat content [22]. This relationship agreed with the greater acetate-to-propionate ratio detected in the rumen of the H-MF group. Under the same dietary condition and stage of lactation, the changes in the acetate-to-propionate ratio in the rumen could be related to differences in the profiles of the microbial community [23].

### 4.2. Relationships between Ruminal Bacteria and Milk Fat Content

The fact that *Firmicutes* and *Bacteroidetes* were the most abundant bacterial phyla regardless of milk fat content in the present study was in line with the result of Pitta et al. using Holstein cows consuming different diets to induce or to recover from milk fat depression [24]. They reported that *Firmicutes* and *Bacteroidetes* together constituted 85 to 86% of the total bacterial sequences in the rumen of cows allocated to different diets. Further, Jami et al. reported that the ratio of *Firmicutes* to *Bacteroidetes* was strongly correlated with milk fat content [25]. The present study supports the view that the ratio of *Firmicutes* to *Bacteroidetes* in the rumen of cows with high milk fat content is greater.

Despite the lack of differences in the richness and diversity of ruminal bacteria as a function of milk fat content divergence, there were some unique differences in terms of relative bacterial abundance. For instance, *Eubacterium_xylanophilum* isolated from the rumen ferments xylan to form formate, acetate, and butyrate [26]. Thus, this indicates that the greater abundance of xylan-fermenting bacteria and xylanase activity in H-MF cows could have led to a better capacity for utilizing xylan in the rumen. Romero et al. reported that the xylanase-rich exogenous enzyme added to the TMR can induce a trend for increased milk fat yield in dairy cows [27]. More recent studies provided strong evidence for a correlation between *Eubacterium_xylanophilum* and host lipid metabolism, e.g., a greater abundance of *Eubacterium_xylanophilum* was observed in high-fat-diet-fed mice [28].

Previous work on diet-induced milk fat depression in dairy cows in which the relative abundance of *Lachnospiraceae*, *Butyrivibrio*, and *Acetitomaculum* in the rumen was associated with low milk fat content agrees with the greater abundance of several bacteria in the L-MF group of this study [24,29,30]. *Ruminococcaceae*, *Butyrivibrio,* and *Lachnospiraceae* are involved in unsaturated fatty acid biohydrogenation and are positively correlated with trans fatty acid intermediates such as *trans*-10, *cis*-12 conjugated linoleic acid, and the *trans*-10 isomer, which could inhibit the de novo synthesis of fatty acids in the mammary gland, consequently decreasing milk fat content [9,24,31]. Our results also agree with previous studies indicating that milk fat content was negatively associated with the relative abundance of *Eubacterium_coprostanoligenes* and *Christensenellaceae_R-7* in the rumen [29,32]. *Lachnoclostridium* and *Lachnospiraceae UCG-006* were reported to suppress the growth of short-chain fatty acid (SCFA)-producing bacteria [33]. Considering that certain SCFAs are milk fat precursors, the greater abundance of *Lachnoclostridium_10* and *Lachnospiraceae_ND3007* in the rumen of L-MF cows may have been related to low milk fat content through the suppression of SCFA-producing bacteria.

### 4.3. Relationship between Ruminal Fungi and Milk Fat Content

The finding by ITS rRNA gene sequencing that *Ascomycota* was a core fungus in the rumen agrees with the report by Guo et al. and Shi et al. [15,34]. This observation contrasts with a recent study reporting that *Neocallimastigomycota* was the most predominant fungal phylum in the rumen of Chinese Holstein cows [20,35]. This discrepancy could be due to differences in nutritional management, housing, and/or stage of lactation across the studies.

*Pyrenophora* is a pathogenic fungus and efficiently produces xylanase, which can degrade the intercellular layer of plant cell walls [36]. This feature agrees with our finding of greater xylanase activity in H-MF cows. *Pyrenophora tritici-repentis* was reported to produce aldehyde dehydrogenase (NAD+), which is involved in the conversion of acetyl-CoA to acetic acid [37]. We speculated that the greater abundance of *Pyrenophora* in H-MF cows would have increased ruminal acetate production, which could then supply it to the mammary gland for milk fat synthesis. Unlike *Pyrenophora*, the H-MF cows had fewer *Caecomyces*, *Wallemia,* and *Pithoascus. Caecomyces* have bulbous rhizoids that can penetrate and expand inside the cellulose matrix, and their numbers were also greater in cows fed a high-forage diet [38]. *Wallemia* is a xerophilic filamentous fungus with the ability to secrete a series of glycosidases, and the abundance of the *Wallemia* genus was positively correlated with valeric acid yield in rumen fluid [39,40].

### 4.4. Relationship between Ruminal Metabolite and Milk Fat Content

Under diet-induced milk fat depression in dairy cows, compared with control cows, the contents of glucose, amino acids, and amines in the ruminal fluid were significantly greater [12]. Unlike this previous study, and despite being fed the same diet, concentrations of lipid and organic acids such as succinic acid, LA, octadecanedioic acid, and lauric acid ethyl ester differed significantly between L-MF and H-MF groups. Several studies have established that milk fat yield can be enhanced with the feeding of fatty acids [41,42,43]. Compared with long-chain fatty acids (LCFA), short- and medium-chain fatty acids (SMCFA) are transported by the portal venous system and reach the liver more rapidly; thus, compared with LCFA, the SMCFA might contribute readily to milk fat synthesis [43]. Kadegowda et al. reported that the most efficiently transferred fatty acids between intestine and milk fat were C14:0 and C12:0 [41]. Thus, we speculate that increasing the content of LA in the rumen had the potential to improve milk fat synthesis. We also observed that bioactive lipids such as FAHFA and 12,13-DiHOME were significantly higher in the rumen of H-MF cows. In non-ruminants, FAHFA was reported to possess anti-inflammatory properties and positively modulate insulin sensitivity and glucose tolerance [44]. 12,13-DiHOME, an epoxide hydration product of linoleic acid, was lower in the plasma of mastitis cows [45].

### 4.5. Association of Metabolites with Microbiota

Ruminal fluid metabolites may be the intermediaries of interactions between the ruminal microbiota and host tissues [46]. Our study revealed that some ruminal fluid metabolites were associated with the differential abundance of microbiota. Succinic acid and LA were correlated with *Prevotellaceae_Ga6A1* and *Pyrenophora*, which agreed with the work on diet-induced milk fat depression in which a positive correlation between the succinic acid content and relative abundance of *Prevotella* in ruminal digesta was reported [12]. LA could alter microbiota populations (except fungi) in the rumen of dairy cows through antiprotozoal or antimicrobial effects, e.g., dietary LA could affect *Prevotella*, *Bacteroides*, and *Enterorhabdus* populations in the rumen [47]. The fungal primer in the above study differed from the primers used in the current trial, which might explain the different responses to the relationship between LA and fungi. Many fatty acids are known to possess antifungal activity. LA exerted strong bioactivity against *Aspergillus niger* and *Fusarium spp*. While linoleic (18:2) and linolenic (18:3) acids exhibited antifungal activity against *Pyrenophora avenae* [48], *Pyrenophora* was negatively correlated with PC (18:2/19:2) in our study. Thus, we speculate that linoleic (18:2), as a hydrolysate of PC (18:2/19:2), could inhibit *Pyrenophora.* The mechanisms controlling the antifungal effects of these lipids could not be discerned at the moment.

### 4.6. LA Increased Intracellular TG Synthesis in BMECs

Lauric acid is a SMCFA composed of 12 carbons and can be isolated from coconut oil and palm kernel oil, which was shown to have various beneficial effects on health [49]. In our study, 50 and 100 μM LA significantly stimulated BMEC proliferation, 200 μM LA was without effect, and the highest dose of LA had an inhibitory effect on BMECs. In comparison, previous studies demonstrated that 100 μM LA enhanced the proliferation of HC11 mouse mammary epithelial cells [49,50]. Inherent differences between ruminant and murine mammary cells may contribute to the different responses. However, the positive effect of LA on TG synthesis was in agreement with the work of Yang et al., demonstrating that LA significantly increased TG content in the HC11 cell supernatant [49]. Overall, 200 μM LA supplementation in BMECs increased TG content not through enhancing the cell number but by altering the mRNA abundance of genes encoding lipogenic enzymes. The coordinated nature of the upregulation, which tended to affect most of the metabolic pathways studied, including fatty acid uptake and activation (*CD36* and *ACSL1*) and TG synthesis (*DGAT1*, *DGAT2*, and *GPAM*), as well as the transcriptional regulation factor (*SREBP1*), supported the involvement of LA as a central regulator of milk fat synthesis (Figure 8F).

The concerted action of CD36 and FABP allows for free fatty acid transport across cell membranes [51,52]. Similar to the effect of oleic acid on HC11 mammary epithelial cells, LA enhanced mRNA expression of *CD36* in BMECs suggesting it allowed for extracellular transport of fatty acids into the cells, thereby providing more substrates for TG synthesis [53]. Consistent with the effect of *CD36* in heart tissue, cardiac TG levels were reduced by ~50% in hearts from tamoxifen-inducible cardiomyocyte-specific *CD36* knockout mice [54].

In BMECs, exogenous fatty acids are activated to fatty acyl-CoA before they can be metabolized or inserted into lipid droplets. *ACSL1* is the predominant acyl-CoA synthetase isoform in bovine mammary tissue and in the present study, LA upregulated *ACSL1* and *DGAT*, which likely contributed to TG synthesis [55]. At least in goat mammary cells, LA can be esterified to a greater extent in the TG fraction (sn-2 or sn-3 position) than other lipid fractions [56]. *DGAT* enzymes catalyze the conjugation of a fatty acyl-CoA to diacylglycerol to form TG and are required to accumulate lipids in multiple tissues [57]. *DGAT* has two isoforms, *DGAT1* and *DGAT2*, but they share no amino acid sequence homology. *DGAT1* is the major isoform expressed in the small intestine, testis, skeletal muscle, and mammary tissue, while *DGAT2* is abundant in the liver and adipose tissue [58]. A previous study demonstrated that over-expression of *DGAT1* and *DGAT2* in mouse liver caused a 2.0- and 2.4-fold increase in TG content, respectively [59]. The knockdown of *DGAT1* decreased TG content in bovine mammary epithelial cells [60]. Thus, the fact that the mRNA expression level of *DGAT1* and *DGAT2* in BMECs treated with LA was 3.90- and 2.0-fold higher than BMECs without LA was consistent with previous data in hepatic tissue demonstrating a preference for *DGAT1* for the use of exogenous fatty acids to synthesize TG [61]. In our study, the expression level of *DGAT1* and *DGAT2* correlated with TG content and also with *SREBP1*. *SREBP1* is a key transcription factor whose activation coincides with copious milk fat synthesis [62]. In goat mammary epithelial cells, *SREBP1* knockdown or overexpression can influence the cellular TG content and the expression level of genes associated with TG synthesis such as *ACSS2*, *LPIN1*, and *DGAT* [63,64]. It is unknown whether *DGAT1* or *DGAT2* is the target gene of *SREBP1*, but *DGAT1* or *DGAT2* inhibition in SUM159 cells prevents *SCAP* (an activator of *SREBP1*) transport from the endoplasmic reticulum to the Golgi [65]. This study also discussed evidence indicating that *DGAT*-*SREBP* regulation is conserved across different cell types.

*SREBP1* positively affects de novo fatty acid synthesis by increasing the expression of *ACSS2*, *ACACA*, and *FASN*, key enzymes in de novo fatty acid synthesis [63]. The reduction in the *FASN* expression level decreased the synthesis of C10:0 and C12:0 in goat mammary glands, a phenomenon that contrasts with our finding of decreased expression of *FASN* in BMECs treated with LA [66]. The most likely explanation for the inhibitory effect of LA on the rate of de novo synthesis of fatty acid was that exogenous LA competed with newly synthesized medium-chain acyl-CoA for the sn-2 and sn-3 positions in the TG backbone [56]. The activity of PPARG could represent an important control point of milk TG synthesis in bovine mammary and adipose cells [67]. It has been reported that *PPARG* significantly promoted lipid synthesis in goat mammary epithelial cells, and its mRNA expression can be activated by LCFA [68,69]. Thus, exogenous LA may alter LCFA uptake, activation, and synthesis, hence, offering ligands for binding and activating *PPARG*. Such an effect may be a reason for the decrease in *PPARG* expression after LA supplementation.

The increase in intracellular TG content was also consistent with the increase in the expression of *GPAM* coding for a protein involved in the first step of TG synthesis [70]. Yu et al. used CRISPR/Cas9 technology to knock out *GPAM* in BMECs and observed that the concentrations of TG, cholesterol, and fatty acids were significantly decreased along with the mRNA expression of *AGPAT6, DGAT1*, and *LPIN1* [71].

## 5. Conclusions

The study demonstrated that the milk fat content of Chinese Holstein cows is associated with microbial profiles and metabolites in the rumen. Compared with L-MF, the H-MF cows had a greater relative abundance of xylan-fermenting bacterium and fungus, suggesting a better capacity for utilizing hemicellulose in the rumen. In contrast, the L-MF cows had a greater abundance of bacteria, which could suppress SCFA-producing bacteria or participate in fatty acid biohydrogenation. The differentially affected ruminal metabolites between the two groups were mainly lipids and organic acids involved in fatty acid biosynthesis, propanoate metabolism, and butanoate metabolism. Furthermore, LA supplementation promoted TG synthesis in BMECs via upregulated fatty acid uptake and activation, as well as lipogenesis.

## Figures and Tables

**Figure 1 animals-14-01493-f001:**
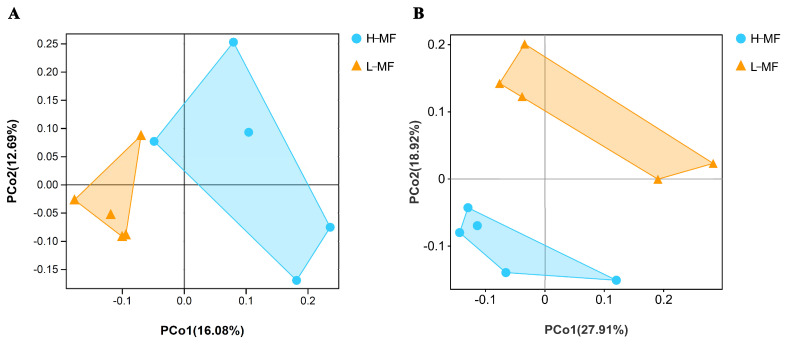
Measure of microbial beta diversity at the genus level. PCoA plot of (**A**) bacterial and (**B**) fungal profiles in ruminal fluid from dairy cows with different milk fat contents. The triangle represents L-MF, and the circle represents H-MF. The percentage variation explained by each principal coordinate is indicated on the axes. The closer the distance between two points, the less the difference in microbial community structure between the two samples. L-MF, low milk fat content group, H-MF, high milk fat content group.

**Figure 2 animals-14-01493-f002:**
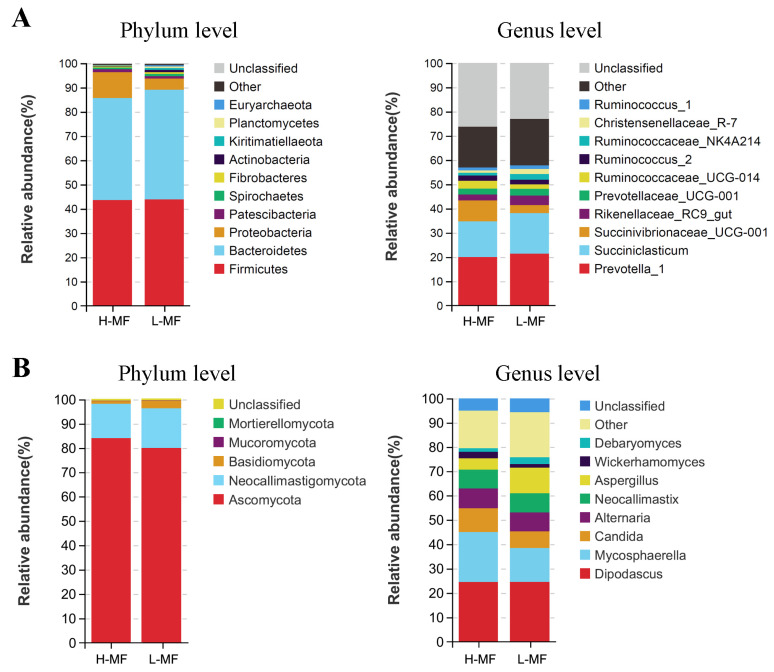
Relative abundances of (**A**) bacteria and (**B**) fungi at the phylum and genus level. Different colors indicate different microorganisms. L-MF, low milk fat content group, H-MF, high milk fat content group.

**Figure 3 animals-14-01493-f003:**
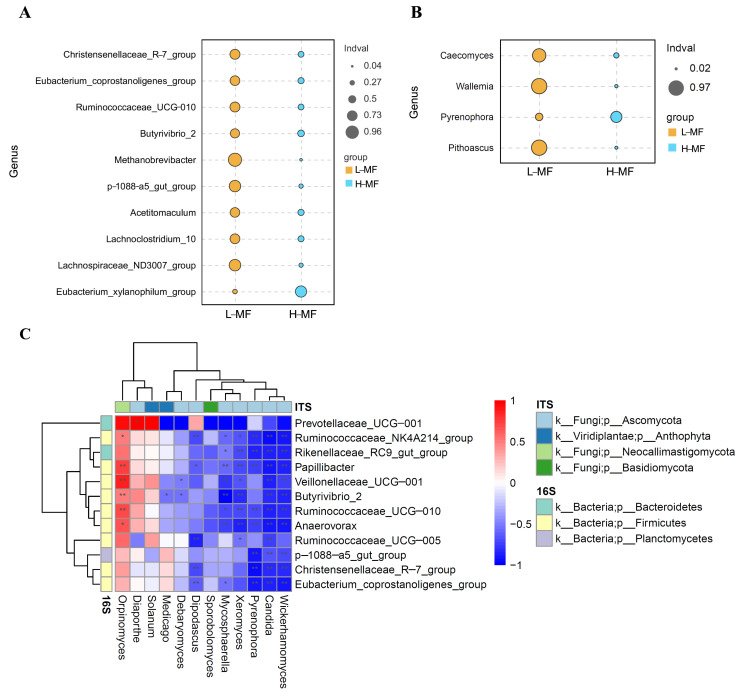
Different (**A**) bacterial and (**B**) fungal genus (*p* < 0.05) in ruminal fluid from dairy cows with different milk fat contents. The yellow circle represents L-MF, and the blue circle represents H-MF. L-MF, low milk fat content group, H-MF, high milk fat content group. (**C**) SparCC heat map depicting the correlations between ruminal bacteria and fungi at the genus level. * *p* < 0.05, ** *p* < 0.01. Red represents a positive correlation, and blue represents a negative correlation. ITS, internal transcribed spacer.

**Figure 4 animals-14-01493-f004:**
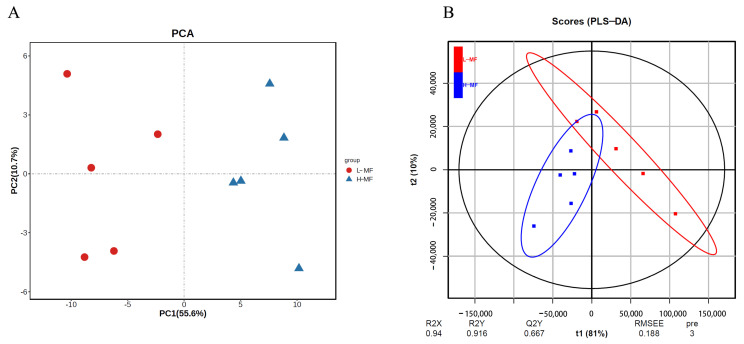
(**A**) PCA and (**B**) PLS-DA score plot based on metabolites data in rumen fluid. The red represents L-MF, and the blue represents H-MF. L-MF, low milk fat content group, H-MF, high milk fat content group.

**Figure 5 animals-14-01493-f005:**
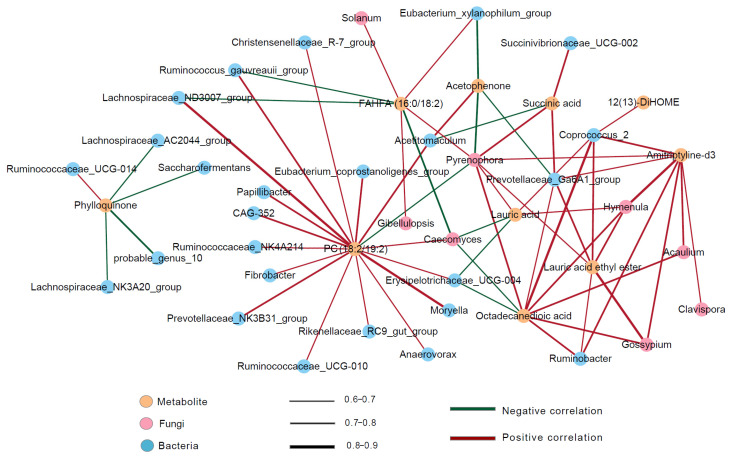
Network showing bacteria and fungi (at the genus level) correlated to different ruminal fluid metabolites (*p* < 0.05 and VIP > 1) by Pearson correlation analysis. The yellow, pink, and blue circles represent metabolites, fungi, and bacteria, respectively. The green and red lines represent negative and positive correlations between metabolites and microbiota. The thickness of the connector line illustrates the strength of the correlation coefficient (0.9 > |r| > 0.6). Example: lauric acid was positively correlated with *Coprococcus_2* (bacteria), *Pyrenophora* (fungi), and *Hymenula* (fungi), and negatively correlated with *Erysipelotrichaceae_UCG-004* (bacteria) and *Caecomyces* (fungi).

**Figure 6 animals-14-01493-f006:**
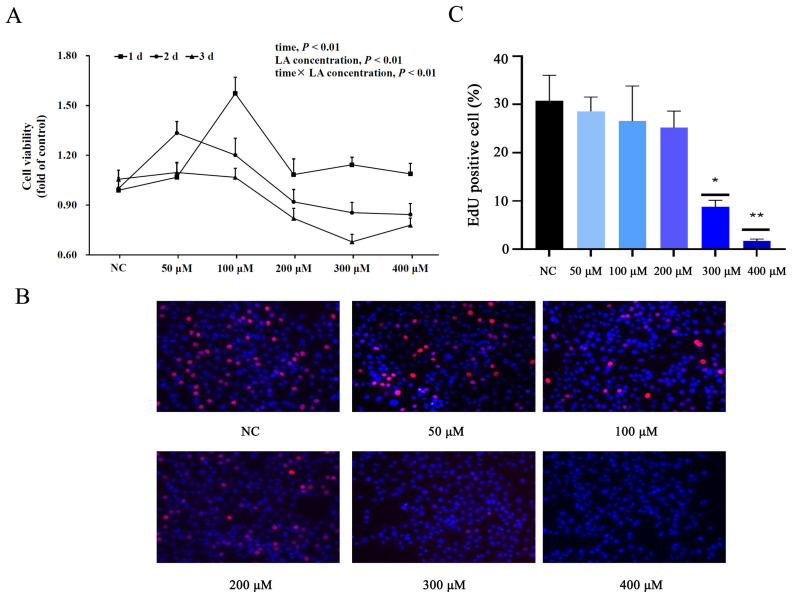
Effect of different concentrations of LA on the cell proliferation of BMECs by using CCK-8 analysis (**A**) and EdU incorporation assay (**B**,**C**). * *p* < 0.05, ** *p* < 0.01.

**Figure 7 animals-14-01493-f007:**
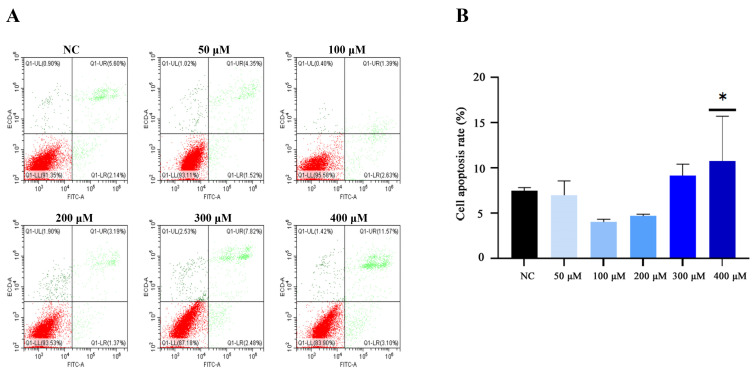
Effect of different concentrations of LA on apoptosis of BMECs by using flow cytometry analysis (**A**) and analysis of cell apoptosis rate in panel A (**B**). Q1-LL represents viable cell, Q1-LR represents early apoptosis of cell, Q1-UR represents late apoptosis of cell, and Q1-UL represents non-viable cell. * *p* < 0.05.

**Figure 8 animals-14-01493-f008:**
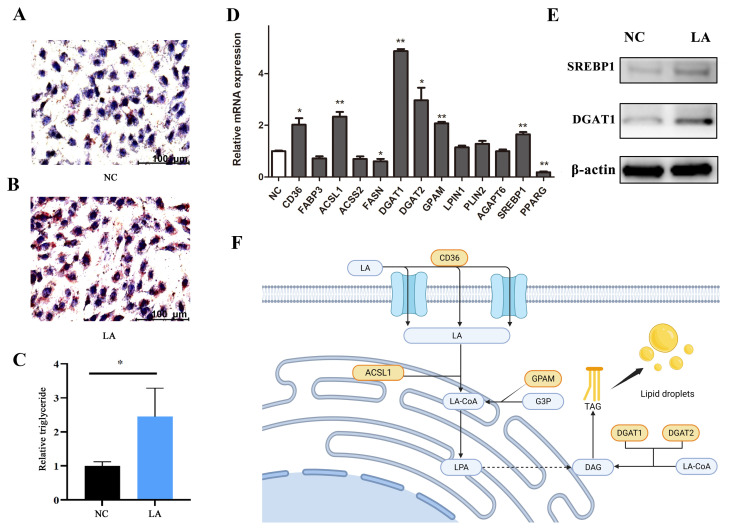
Effect of LA on lipid metabolism in BMECs. (**A**) Oil red O result of NC, the scale bar = 100 μm, (**B**) Oil red O result of 200 μM LA-treated BMECs, (**C**) Relative TG content, (**D**) The relative mRNA expression level of genes related to lipid metabolism after 200 μM LA treatment in BMECs, (**E**) Western blotting analysis of the level of protein related to lipid metabolism after 200 μM LA treatment in BMECs, (**F**) Graphical representation of the effect of LA on TG synthesis in BMECs, this image was created with bioRender.com (accessed on 30 November 2023). * *p* < 0.05, ** *p* < 0.01.

**Table 1 animals-14-01493-t001:** Milk yield and composition of L-MF and H-MF groups.

Items	Groups		*p*-Value
	L-MF	H-MF	
Parity	1.80 ± 0.37	1.80 ± 0.37	1.00
Lactation days	257.80 ± 55.90	218.80 ± 77.98	0.70
Milk yield (kg/d)	28.54 ± 1.90	27.96 ± 5.34	0.92
Dry matter intake (kg/d)	22.50 ± 1.85	21.57 ± 1.80	0.11
Average milk fat content (%)	3.51 ± 0.12	4.55 ± 0.19	<0.01
Milk fat content (%)	3.60 ± 0.12	5.82 ± 0.41	<0.01
Milk protein content (%)	3.29 ± 0.16	3.92 ± 0.50	0.27
Milk lactose (%)	4.90 ± 0.11	4.81 ± 0.10	0.59
Whole milk solids content (%)	12.98 ± 0.58	16.04 ± 0.86	0.02
Solids-not-fat content (%)	8.93 ± 0.22	9.50 ± 0.51	0.34
MUN content (mg dL^−1^)	12.02 ± 0.73	12.95 ± 1.36	0.57
SCS	1.00 ± 0.63	1.60 ± 0.68	0.54
Milk fat LA concentration (mg per 100 g milk)	156.48 ± 17.29	217.13 ± 13.21	0.02

L-MF, low milk fat content group, H-MF, high milk fat content group, MUN, milk urea nitrogen, SCS, somatic cell score.

**Table 2 animals-14-01493-t002:** Comparative analysis of VFA concentrations and digestive enzyme activities in the ruminal fluid of dairy cows with different milk fat contents.

Items	Groups		*p*-Value
	L-MF	H-MF	
pH	6.51 ± 0.16	6.64 ± 0.13	0.52
Acetate (mmol L^−1^)	61.23 ± 7.34	67.58 ± 2.25	0.43
Propionate (mmol L^−1^)	19.70 ± 2.32	16.91 ± 1.73	0.36
Isobutyrate (mmol L^−1^)	0.41 ± 0.06	0.46 ± 0.02	0.40
Butyrate (mmol L^−1^)	7.99 ± 1.47	8.62 ± 1.02	0.73
Isovalerate (mmol L^−1^)	0.86 ± 0.12	0.92 ± 0.04	0.63
Valerate (mmol L^−1^)	0.83 ± 0.11	0.82 ± 0.09	0.91
Acetate: Propionate	3.13 ± 0.30	4.10 ± 0.25	0.04
Total VFA (mmol L^−1^)	91.02 ± 10.37	95.31 ± 4.94	0.72
Cellulase (mU L^−1^)	39.64 ± 1.12	38.40 ± 0.92	0.42
Xylanase (U L^−1^)	30.70 ± 1.08	36.69 ± 0.66	0.04
Lipase (U L^−1^)	12.17 ± 0.70	4.05 ± 0.82	<0.01
Pectinase (U L^−1^)	63.04 ± 4.20	60.51 ± 1.54	0.59
Proteinase (U mL^−1^)	304.25 ± 22.62	261.80 ± 30.02	0.04
Amylase (U mL^−1^)	5.89 ± 0.68	6.72 ± 0.66	0.08

L-MF, low milk fat content group, H-MF, high milk fat content group, VFA, volatile fatty acids.

**Table 3 animals-14-01493-t003:** α diversity index of ruminal bacterial and fungal communities.

Index	16S		*p*-Value	ITS		*p*-Value
	L-MF	H-MF		L-MF	H-MF	
Sob	1690.20 ± 31.90	1595.20 ± 45.40	0.13	232.20 ± 15.12	217.80 ± 16.21	0.18
Shannon	7.91 ± 0.21	7.14 ± 0.47	0.18	3.03 ± 0.58	2.98 ± 0.40	0.88
Simpson	0.97 ± 0.00	0.93 ± 0.03	0.20	0.69 ± 0.12	0.69 ± 0.10	0.96
Chao1	1753.91 ± 32.10	1665.04 ± 41.68	0.13	272.62 ± 18.35	249.53 ± 20.14	0.09
Ace	1833.35 ± 29.18	1742.80 ± 43.24	0.12	273.34 ± 20.10	251.94 ± 12.57	0.08

L-MF, low milk fat content group, H-MF, high milk fat content group, ITS, internal transcribed spacer.

**Table 4 animals-14-01493-t004:** Differential metabolites in ruminal fluid from dairy cows with different milk fat contents.

Metabolites	Formula	Retention Time (min)	*m*/*z*	log_2_FC	*p*-Value	VIP
Lauric acid ethyl ester	C_14_H_28_O_2_	14.49	227.20	0.95	0.01	6.18
Phosphatidylcholine (18:2/19:2)	C_45_H_82_NO_8_P	16.35	778.57	−1.44	0.03	3.60
Octadecanedioic acid	C_18_H_34_O_4_	14.49	315.25	0.57	0.03	2.76
Lauric acid	C_12_H_24_O_2_	13.82	199.17	0.59	0.00	2.04
12(13)-DiHOME	C_18_H_34_O_4_	12.59	313.24	0.79	0.03	1.77
Acetophenone	C_8_H_8_O	7.91	121.06	−0.80	0.03	1.26
Succinic acid	C_4_H_6_O_4_	1.23	135.03	0.73	0.01	1.20
4-(Diethylamino) salicylaldehyde	C_11_H_15_NO_2_	8.66	194.12	−0.99	0.02	1.10
FAHFA (16:0/18:2)	C_34_H_62_O_4_	15.04	533.46	0.31	0.04	1.04
Phylloquinone	C_31_H_46_O_2_	16.20	451.36	1.30	0.03	1.01
Amitriptyline-d3	C_20_H_23_N	14.22	281.21	0.84	0.04	1.01

FC, fold change in the metabolite concentration (H-MF/L-MF), VIP = variable importance in projection, 12(13)-DiHOME, 12,13-dihydroxy-9(Z)-octadecenoic acid, and FAHFA, branched fatty acid esters of hydroxy fatty acids.

## Data Availability

All data generated and analyzed during this study are included in this published article. Raw data supporting the findings of this study are available from the corresponding author on request.

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
