# Peer review of "Potential Role of Lauric Acid in Milk Fat Synthesis in Chinese Holstein Cows Based on Integrated Analysis of Ruminal Microbiome and Metabolome"

_animals, 2024, doi:10.3390/ani14101493_

Round 1
Reviewer 1 Report
Comments and Suggestions for Authors
A kindly suggestion. For me it would be important to identify the genetic history of each cow used in this research, because if the cows were similar in age, calving rate, milk production, consuming the same diet, it is possible that the original parents of each (mainly the father ) could have had different parents and inherited certain genes to their daughters to present these differences in rumen metabolism and milk fat synthesis.
Author Response
A kindly suggestion. For me it would be important to identify the genetic history of each cow used in this research, because if the cows were similar in age, calving rate, milk production, consuming the same diet, it is possible that the original parents of each (mainly the father) could have had different parents and inherited certain genes to their daughters to present these differences in rumen metabolism and milk fat synthesis.
AU: Thank you for your comments, and we have added the genetic history of each cow used in this research in Line 115. After confirmed the records in Wanhe dairy farm, these cows used in our research are all half sibs, they have the same father.
Reviewer 2 Report
Comments and Suggestions for Authors
Respected Authors,
Thank you very much for the interest reading of your manuscript. I appreciate your work that you put into your manuscript.
I have som specific comments to your submission:
Because the line numbering is missing in your manuscript i will be describing my comments as I found them in paragraphs.
In Abstract - I recommend to more specify the goal of your experiment after the first sentence.
2.1 Cows, experimental design and diets.
I think it could be beneficial to have control group, with cow with average milk fat content. My opinion is that is very difficult to determine the observed differences in the rumen bacteria and metabolites are due to the milk fat content or other factors.
2.2 Sample collection
How many days and how (fridge, type, manufacturer) were milks samples stored? I recommend that the samples were collected via esophagus and add the type of the hose/tube and sunction method (electric?syringe?) How much of the sample was needed for the pH rumen fluid analysis? How was possible to not contamine the sample? Who was the collector of the samples? veterinarian?
Table 1 - there are missing abbreviations under the table
Table 2 - there are missing abbreviations under the table
Table 3 - there are missing abbreviations under the table
Table 4 - there are missing abbreviations under the table
Best Regards,
Reviewer
Reviewer 3 Report
Comments and Suggestions for Authors
In the study, the authors examined a Integrated analysis of ruminal microbiome and metabolome reveals a potential role for lauric acid in milk fat synthesis in Chinese Holstein cows.
However, the following observations can be made.
1. Title of the manuscript. Very long and incomprehensible. I propose to change it and make it more clear. For example, "Potential role of lauric acid in milk fat synthesis in Chinese Holstein cows based on Integrated analysis of ruminal microbiome and metabolome".
2. In your abstract you write that “Microbiome sequencing revealed 10 bacteria and 4 fungi...”. It is not right. You have identified 10 types of bacteria and 4 types of fungi. Probably so!
3. Figures 2 and 4 are missing from the manuscript.
4. Conclusion. It is not correct to write Holstein cows. KU Vas are Holstein cows of Chinese selection (Chinese Holstein cows).
Comments on the Quality of English Language
The manuscript was prepared in English.
However, there are minor typos in the text.
I recommend editorial changes to the English language of the manuscript.
Reviewer 4 Report
Comments and Suggestions for Authors
Congratulations to the authors for the excellent manuscript presented here. A work with great originality, new information for science and which should be published.
1) the summary is excellent, very well written, excellent English.
2) It was also very clear in the introduction section the knowledge gap that the authors sought to investigate, but at the end of this section, the authors could make it clear what your hypothesis would be.
3) why did the authors select only 10 animals for this experiment? It was not clear what criteria were used to select these animals from a herd of more than 500 animals.
4) Why didn't the authors use the NRC 2021? It is much more up to date and there would certainly be a difference between the diets.
5) more details on how the statistical analysis was carried out; starting with a normality test, as it is important to know whether the data had a normal distribution, for example.
6) in in vitro tests with cells, why was the interaction treatment x moment/hours not evaluated in the statistical analysis?
7) Table 1 - almost 40 days is the average interval between the two groups. Could it be that this alone did not influence your results?
8) Figures 1 and 2 could be better centralized in the file; Furthermore, the subtitles must be improved; leaving it self-explanatory.
9) the caption of figure 5 must make it clear what the reader should be when observing the figure; If necessary, draw attention to the main points.
10) Discussion and conclusion sections were good; but even so, I remind researchers that the conclusion must respond to the objective; with this one now, she does many more than that. Reflect
I believe that after these adjustments, the paper can be accepted for publication.
Round 2
Reviewer 2 Report
Comments and Suggestions for Authors
Respected Author,
Thank you very much for accepting my suggestion and responding to my comments.
I suggest to accept your submission in present form.
Best Regards,
Reviewer
Reviewer 4 Report
Comments and Suggestions for Authors
Authors, congratulations on the excellent work. It was reviewed and the questionnaires were answered appropriately. Certain to be a good contribution to science. I recommend accepting it.